# Chicken Interferon-Alpha and -Lambda Exhibit Antiviral Effects against Fowl Adenovirus Serotype 4 in Leghorn Male Hepatocellular Cells

**DOI:** 10.3390/ijms25031681

**Published:** 2024-01-30

**Authors:** Jinyu Lai, Xingchen He, Rongjie Zhang, Limei Zhang, Libin Chen, Fengping He, Lei Li, Liangyu Yang, Tao Ren, Bin Xiang

**Affiliations:** 1College of Veterinary Medicine, Yunnan Agricultural University, Kunming 650201, China2018018@ynau.edu.cn (L.Z.); 1993009@ynau.edu.cn (L.Y.); 2College of Veterinary Medicine, South China Agricultural University, Guangzhou 510642, China; 3Center for Poultry Disease Control and Prevention, Yunnan Agricultural University, Kunming 650201, China

**Keywords:** antiviral activity, fowl adenovirus serotype 4, hydropericardium hepatitis syndrome, interferon, interferon-stimulated genes

## Abstract

Hydropericardium hepatitis syndrome (HHS) is primarily caused by fowl adenovirus serotype 4 (FAdV-4), causing high mortality in chickens. Although vaccination strategies against FAdV-4 have been adopted, HHS still occurs sporadically. Furthermore, no effective drugs are available for controlling FAdV-4 infection. However, type I and III interferon (IFN) are crucial therapeutic agents against viral infection. The following experiments were conducted to investigate the inhibitory effect of chicken IFN against FadV-4. We expressed recombinant chicken type I IFN-α (ChIFN-α) and type III IFN-λ (ChIFN-λ) in *Escherichia coli* and systemically investigated their antiviral activity against FAdV-4 infection in Leghorn male hepatocellular (LMH) cells. ChIFN-α and ChIFN-λ dose dependently inhibited FAdV-4 replication in LMH cells. Compared with ChIFN-λ, ChIFN-α more significantly inhibited viral genome transcription but less significantly suppressed FAdV-4 release. ChIFN-α- and ChIFN-λ-induced IFN-stimulated gene (ISG) expression, such as *PKR*, *ZAP*, *IRF7*, *MX1*, *Viperin*, *IFIT5*, *OASL*, and *IFI6*, in LMH cells; however, ChIFN-α induced a stronger expression level than ChIFN-λ. Thus, our data revealed that ChIFN-α and ChIFN-λ might trigger different ISG expression levels, inhibiting FAdV-4 replication via different steps of the FAdV-4 lifecycle, which furthers the potential applications of IFN antiviral drugs in chickens.

## 1. Introduction

An outbreak of fowl adenoviruses occurred in China in 2015 and rapidly disseminated across the country [1]. The subgroup C fowl adenovirus serotype 4 (FAdV-4) within group I was the most destructive among the fowl adenoviruses involved in the outbreak, responsible for inducing hydropericardium syndrome (HHS) [2,3]. Furthermore, the immunosuppressive effects induced by fowl adenoviruses contribute to secondary infections with other pathogens, increasing mortality in nearly 100% of the affected population [4,5,6]. Furthermore, FAdV-4 can be cross-transmitted between various hosts, such as laying hens, broilers, ducks, mandarin ducks, geese, and wild birds [7,8,9]. FAdV-4 is transmitted through the fecal–oral route and vertically via breeding eggs, posing challenges in adenovirus prevention and resistance [10]. Vaccination strategies against FAdV-4 have been adopted; however, HHS still occurs sporadically. In addition, immunosuppression in chickens after infection with various pathogens frequently leads to suboptimal immunization outcomes with commercial vaccines [11]. Our recent study focused on the epidemiology of fowl adenoviruses and revealed the presence of FAdV-4 in healthy flocks [12]. Thus, developing anti-FAdV-4 drugs could be a valuable adjunct to vaccine immunization, expanding the range of preventive and control measures available for managing HHS.

IFNs were categorized into three groups, type I, type II, and type III IFNs, based on their gene sequence, molecular structure, chromosomal localization, and receptor specificity [13]. The chicken interferons (IFNs) IFN-α, IFN-β, IFN-κ, IFN-ω, IFN-ζ, and IFN-τ were successively characterized in 2000, and ChIFN-λ was genetically engineered for expression in 2008 [14,15,16]. Previous studies revealed that the infection of chickens with FAdV-4 induces a strong innate immune response, including the upregulation of IFN expression [17,18,19]. Li et al. revealed that FAdV-4 induces cellular pathways in chickens to produce IFNs and antigen-presenting molecules (MHCI/II) [20]. However, research investigating the impact of IFNs on adenoviruses is lacking. Previous investigations into the antiviral properties of IFNs predominantly concentrated on type I IFNs. The presence of type I IFNs in fowl species were elucidated by characterizing its two primary members, IFN-α and IFN-β [21], and IFN-α exhibits more robust antiviral efficacy than IFN-β [22]. Nevertheless, emerging evidence suggests that IFN-λ is essential to mucosal immunity against viral pathogens [23]. Consequently, IFN-α was selected as the focus of this study. Furthermore, as a newly identified IFN, whether IFN-λ exhibits anti-FAdV-4 activity is unknown.

IFNs are versatile antiviral medications which demonstrated their efficacy in suppressing various viruses, such as encephalomyocarditis, lymphocytic choriomeningitis, herpes simplex virus type 2, and hepatitis B viruses [24,25]. Type I and type III IFNs also have inhibitory properties against severe acute respiratory syndrome-related coronavirus 2 (SARS-CoV-2), suggesting their potential as therapeutic options for SARS-CoV-2 infections. Recombinant type I IFN can effectively hinder the infection of various viruses in avian species, such as fowl leukemia, Marek’s, highly pathogenic fowl influenza, and infectious bronchitis [26,27,28,29]. Nevertheless, the effects of type I and type III IFNs differ significantly among different viruses. For example, type III IFNs, specifically IFN-λ, exhibit greater efficacy against porcine epidemic diarrhea virus (PEDV) than IFN-α [30].

Type I and type III IFNs bind to different receptors on the cell membrane, with chicken type I IFNs binding to IFNAR1 and IFNAR2 and type III IFNs binding to IFNLR1 and IL10R2 [31,32]. These receptors are distributed in different abundances in different cells, and the receptors for type I IFNs are primarily distributed in fibroblasts. In contrast, the receptors for type III IFNs are mainly distributed in epithelial cells [33]. Leghorn male hepatocellular (LMH) cells were characterized and established in 1987 and belong to the epithelial cell line, which can be used to study FAdVs [34,35]. IFN induces many IFN-stimulated genes (ISGs) crucial for the antiviral response, and this induction relies on the Janus kinase/signal transducer and activator of the transcription signaling pathway [36]. Oligoadenylate synthetase (OAS) is a secondary messenger that detects foreign RNA, exerting antiviral effects [37]. Viperin hinders viral replication by impacting cellular metabolism and mitochondrial function [38]. Mx dynamin-like GTPases are crucial as antiviral effectors in type I and III IFN systems. These proteins exert their inhibitory effects on various viruses by impeding the initial stages of viral replication [39]. Initially identified as an ISG, IFI6 was initially observed to localize in mitochondria; however, subsequent investigations revealed its presence in the endoplasmic reticulum (ER). IFI6 serves a prophylactic function by safeguarding uninfected cells against virus-induced invagination formation in the ER membrane [40]. The IFN-induced protein with tetratricopeptide repeats (IFIT) family plays a crucial role in various processes that counteract viral infection, primarily by modulating translation, which consequently impedes viral replication [41]. Although the limiting effect of chicken IFN on various avian pathogens was demonstrated, the effect of chicken IFNs on FAdV-4 was not reported.

In this study, chicken type I IFN-α (ChIFN-α) and type III IFN-λ (ChIFN-λ) were prokaryotically expressed, and their ability to inhibit FAdV-4 in LMH cells was compared.

## 2. Results

### 2.1. ChIFN-α and ChIFN-λ Structure Prediction

ChIFN-α and ChIFN-λ possess signal peptides at the N-terminus. Consequently, based on the signal peptide prediction results, we amplified ChIFN-α (residues 32–193) and ChIFN-λ (residues 24–186) using polymerase chain reaction (PCR) and cloned the two genes into the pET-32α vector respectively, resulting in the creation of pET-32α-ChIFN-α and pET-32α-ChIFN-λ plasmids, both of which were fused with a His-Tag at the N-terminus.

A three-dimensional reconstruction was performed using the SWISS-MODEL online server (https://swissmodel.expasy.org/; accessed on 19 May 2023) to determine the structures of recombinant ChIFN-α and ChIFN-λ. Recombinant ChIFN-α and ChIFN-λ have similar structures, comprising five α-helices (Figure 1). The five α-helices of recombinant ChIFN-α are arranged in a barrel-like configuration, whereas the five α-helices of recombinant ChIFN-λ exhibit an irregular distribution when aggregated.

### 2.2. ChIFN-α and ChIFN-λ Expression and Purification

The recombinant ChIFN-α and ChIFN-λ expression was further verified using Western blotting (Figure 2A). The expression strain *E. coli BL21 (DE3)* was used to obtain natural recombinant ChIFN-α and ChIFN-λ, and induction was achieved using isopropyl β-d-1-thiogalactopyranoside. The expression was validated, demonstrating the expression of recombinant ChIFN-α and ChIFN-λ in inclusion bodies (Figure 2B,C). Following nickel column purification, the protein purity exceeded 90% after dialysis (Figure 2B,C). The concentration of ChIFN-α was 0.0565 mg/mL, and the ChIFN-λ concentration was 0.165 mg/mL after ultrafiltration.

### 2.3. Assessing the Antiviral Efficacy of ChIFN-α and ChIFN-λ

Both CHIFN-α and CHIFN-λ showed activity against Porcine vesicular stomatitis virus (VSV). The specific activity of ChIFN-α in LMH cells was 3.96 × 10^4^ UI/mg, whereas that of ChIFN-λ was 2.3 × 10^4^ UI/mg. These findings demonstrated a dose-dependent relationship between the anti-FAdV-4 activity of recombinant ChIFN-α and ChIFN-λ (Figure 3). Considerable inhibition of FAdV-4 hexon expression was observed with the administration of 100 ng of recombinant ChIFN-α and ChIFN-λ. Furthermore, the inhibitory effect on FAdV-4 hexon expression increased gradually with the concentration of the treatments (Figure 3A,C). However, this dose-dependent inhibitory effect became non-significant at 48 h post-infection (hpi) (Figure 3B,D).

### 2.4. ChIFN-α and ChIFN-λ Pretreatments Restrict FAdV-4 within a Specific Temporal Horizon

Various experiments, including Western blotting, quantitative real-time PCR, and viral titration, were conducted in LMH cells to elucidate the impact of recombinant ChIFN-α and ChIFN-λ on FAdV-4 restriction.

The inhibitory effect of recombinant ChIFN-α and ChIFN-λ on hexon expression in FAdV-4 was significant within 48 hpi (Figure 4A,B). Furthermore, when LMH cells were pretreated with recombinant ChIFN-α and ChIFN-λ, hexon expression was significantly inhibited 24 h after inoculation with FAdV-4. However, the suppressive effect of recombinant ChIFN-α and ChIFN-λ on FAdV-4 gradually diminished as the duration of viral infection increased.

The transcription levels of LMH cells infected with FAdV-4 decreased upon recombinant ChIFN-α and ChIFN-λ treatment. Specifically, the hexon transcription of FAdV-4 was significantly reduced in LMH cells pretreated with recombinant ChIFN-α and ChIFN-λ compared to the control group. This significant reduction was observed at 24, 36, and 48 h post-treatment (Figure 4C).

Pretreating LMH cells with ChIFN-α and ChIFN-λ significantly reduced viral release (Figure 4D). At 12 hpi after inoculation with FAdV-4, the viral titer of the control group reached 10^1.667^, the viral titer of the ChIFN-α-treated group was 10^1.287^, and the viral titer of the ChIFN-λ-treated group was 0. ChIFN-λ exhibited a substantial inhibitory effect on FAdV-4 release during the initial stages of viral infection, resulting in a 2-log10 reduction in viral titer within 24 hpi. These findings suggest that ChIFN-λ plays a crucial role in impeding the early replication process of FAdV-4. However, ChIFN-α exhibited no significant FAdV-4 release inhibition during the initial phase of FAdV-4 infection. Nonetheless, ChIFN-α demonstrated a more pronounced impact during the later stages of FAdV-4 replication, specifically between 36 and 72 hpi.

### 2.5. ChIFN-α and ChIFN-λ Induce ISG Expression in LMH Cells

LMH cells were exposed to 1000 UI of ChIFN-α or ChIFN-λ for 12 h to investigate their antiviral effects. ChIFN-α and ChIFN-λ had a stimulatory effect on the mRNA levels of ISGs, including *IFITM3*, *PKR*, *ZAP*, *IRF7*, *MX1*, *Viperin*, *IFIT5*, *OASL*, and *IFI6* (Figure 5). The increase in *PKR*, *ZAP*, *IRF7*, *MX1*, *ASL*, *IFI6*, *IFIT5*, and *Viperin* upregulation was significant (ChIFN-α treatment group: 4.5-, 4.5-, 6.9-, 12-,192.3-, 187.1-, 350.6-, 716-, and 771.3-fold, respectively; ChIFN-λ treatment group: 1.1-,1.5-, 2.8-, 3.8-, 34.9-, 30.8-, 191.8-, 160-, 40.7-fold, respectively). Among the ISGs examined, ChIFN-α induced significantly higher ISG levels than ChIFN-α.

## 3. Discussion

FAdV-4 is a significant emerging pathogen, causing harm to poultry and infecting a diverse array of fowl species [42]. Numerous investigations substantiated the escalating prevalence of FAdV-4 infection in Chinese poultry, with evidence of cross-species transmission further complicating prevention and control efforts [7]. Consequently, other prevention and control measures should be conducted to supplement the deficiency of vaccine immunization.

Considering the specificities of the poultry industry, namely the imperative to minimize drug production costs, we opted to use cost-effective prokaryotic expression to generate recombinant ChIFN-α and ChIFN-λ. However, the complex structure of ChIFN-α and ChIFN-λ resulted in their expression as inclusion bodies during prokaryotic expression, rendering them biologically inactive. Therefore, the inclusion bodies were purified under denaturing conditions to address this issue, and optimal complexity conditions were determined to facilitate the folding of ChIFN-α and ChIFN-λ into their native conformations.

In this study, pretreatment with ChIFN-α and ChIFN-λ caused substantial FAdV-4 inhibition within 24 hpi. However, this inhibitory effect exhibited a significant decline after 48 hpi. This phenomenon could be attributed to the gradual reduction in biological activity of IFNs as treatment duration increases, particularly under elevated temperatures.

ChIFN-α and ChIFN-λ demonstrated significant FAdV-4 release inhibition within 72 hpi; however, discernible differences in their antiviral capabilities were observed. The biological activity of ChIFN-α was significantly higher than that of ChIFN-λ; however, ChIFN-λ was more effective than ChIFN-α in inhibiting viral release. Therefore, ChIFN-λ exhibited a greater ability to inhibit FAdV-4 release, particularly during the early stages of viral infection (12–24 hpi), compared to ChIFN-α. Numerous studies extensively compared the antiviral abilities of type I and III IFNs. Type I IFNs exhibit significantly superior antiviral efficacy against Crimean–Congo hemorrhagic fever virus and SARS-CoV-2 compared to type III IFNs [43,44]. However, type III IFNs also exert a markedly stronger restriction effect on PEDV than type I IFNs in intestinal epithelial cell lines [30]. Hence, the antiviral capacities of type I and III IFNs vary depending on the specific virus and cell line. Type I IFN receptors (IFNAR1 and IFNAR2) are primarily expressed in fibroblasts, whereas type III IFN receptors (IFNLR1 and IL10R2) are predominantly expressed in epithelial cells or tissues abundant in epithelial cells [45,46]. Previous studies that systematically examined the induction of ChIFN-λ-stimulated genes in LMH cells identified a total of 421 type III ISGs [47]. In contrast, only 115 ISGs of ChIFN-λ were identified in DF-1 cells [48]. This suggests that differences in receptor expression in different cell lines may lead to differences in ISG induction. Variations in receptor expression levels and the affinity for receptor binding impact the antiviral efficacy of IFNs. For instance, despite sharing the same receptor, IFN-α and IFN-β display distinct levels of antiviral activity, which is attributed to the stronger affinity of IFN-α for IFNAR1 and IFNAR2 [49]. Consequently, the dissimilarity in antiviral resistance between ChIFN-α and ChIFN-λ may be associated with the distribution of receptors and their respective affinities. ChIFN-α induced higher ISG levels than ChIFN-λ; however, excessive upregulation does not necessarily benefit the antiviral response of the host. This disparity in ISG expression levels may contribute to the stronger inhibitory effect of ChIFN-λ compared to ChIFN-α during the initial phase of FAdV-4 infection.

Many ISGs can participate in diverse viral infection processes. For instance, certain members of the *IFITM* family can hinder viral membrane fusion, impeding viral invasion [50]. Additionally, ZC3HAV1 (*ZAP*) exhibited efficacy in suppressing viral mRNA translation in mammals while concurrently preserving the integrity of host mRNAs [51]. Protein kinase R also plays a crucial role in attenuating viral mRNA translation [52], and mammalian piperin impedes viral replication and restricts viral outgrowth [53]. ISG expression was detected after ChIFN-α and ChIFN-λ treatment to investigate the mechanism of ChIFN-α and ChIFN-λ FAdV-4 inhibition. ChIFN-α and ChIFN-λ promoted the upregulation of mRNAs associated with ISGs, including *PKR*, *ZAP*, *IRF7*, *MX1*, *Viperin*, *IFIT5*, *OASL*, and *IFI6*. *PKR*, *ZAP*, *IRF7 MX1*, *Viperin*, *IFIT5*, *OASL*, and *IFI6* upregulation exhibited significant increases. However, ChIFN-α-induced transcriptional upregulation was significantly higher in all ISGs compared with the ChIFN-λ treatment group.

The present study demonstrates the pronounced inhibitory effects of ChIFN-α and ChIFN-λ on FAdV-4. These effects include suppressing viral transcription in LMH cells, restricting viral protein synthesis, and limiting viral release. These findings strongly indicate that ChIFN-α- and ChIFN-λ-induced ISG expression play a pivotal role in various stages of viral infection. Further studies are underway to identify the effective host restriction factors of FAdV-4 with transcriptome sequencing. In relation to practical implementation, we posit that it holds potential as a therapeutic approach for addressing the immune prevention and control insufficiency associated with FAdV-4, particularly within the precious chicken population. Nevertheless, regarding the precise execution strategy, we contend that additional comprehensive experimentation is imperative. This aspect constitutes our research team’s primary area of interest, which we aim to explore further in subsequent investigations.

## 4. Materials and Methods

### 4.1. Cells, Viruses, and Antibodies

LMH cells were obtained from the American Type Culture Collection (Manassas, VA, USA). Cells were cultured in Dulbecco’s Modified Eagle’s Medium/F12 (Gibco, Waltham, MA, USA) supplemented with 10% fetal bovine serum (Gibco, Waltham, MA, USA) in a 5% CO_2_ incubator at 37 °C. The FAdV-4 KM strain was isolated and preserved in our laboratory and propagated in LMH cells. Vesicular stomatitis virus was obtained from the Department of Avian Diseases, South China Agricultural University. Prof. Shijun J. Zheng from the China Agricultural University kindly donated the hexon monoclonal FAdV-4 antibody [54]. His-Tag Monoclonal antibody (66005-1-Ig) and GAPDH monoclonal rabbit antibody (60004-1-Ig) were purchased from Wuhan Proteintech Group (Proteintech, Wuhan, China), and horseradish peroxidase-labeled goat anti-rabbit antibody (CW0103) and goat anti-mouse antibody (CW0102) were purchased from CWBIO (CWBIO, Beijing, China).

### 4.2. ChIFN-α and ChIFN-λ Structure Prediction and Epression

Signal peptide prediction was performed via the SignaIP website using the ChIFN-α (assembly: EU937528.1) and ChIFN-λ (assembly: EF587763) sequences published on the National Library of Medicine website. The three-dimensional models of the ChIFN-α and ChIFN-λ proteins were reconstructed using the SWISS-MODEL online server (https://swissmodel.expasy.org/; accessed on 19 May 2023). ChIFN-α and ChIFN-λ were amplified using polymerase chain reaction (PCR) from chicken embryo fibroblasts. The primers used during PCR are presented in Table 1. The amplified gene was cloned into the pET32α vector (Novagen; EMD Millipore, Billerica, MA, USA) and expressed in *Escherichia coli BL21 (DE3)*. The specific procedure involved transforming the plasmid into *BL21 (DE3)* and subsequently culturing the *BL21 (DE3)* at 37 °C and 220 rpm until its optical density at 600 nm (OD600) reached 0.6~0.8. Following this, 1 mM isopropyl-β-d-thiogalactoside (IPTG) was introduced for induction, and the culture was sustained for a duration of 10 h. Subsequently, the bacterial solution was collected.

### 4.3. IFN Purification and Renaturation

After harvesting the expression strains, the organisms were lysed using an ultrasonic crusher (Jinxing, Shanghai, China), the inclusion bodies were lysed with 8 M urea, and ChIFN-α and ChIFN-λ were purified using the His-Tagged Protein Purification Kit (CWBIO, Beijing, China). The protein was renatured by gradually removing urea from the purified protein through dialysis. The synthesized proteins were confirmed through 15% sodium dodecyl-sulfate polyacrylamide gel electrophoresis (SDS-PAGE) analysis with Kaumas Brilliant Blue staining and Western blotting with anti-His antibody (1:8000 dilution with Tris-buffered saline [TBS]).

### 4.4. Antiviral Assay

The biological activities of recombinant ChIFN-α and ChIFN-λ were measured using VSV according to previously described methods [55]. The biological activity of ChIFN-α and ChIFN-λ in LMH was analyzed by inhibiting FAdV-4-induced LMH cytopathic effects as described previously [47]. Recombinant IFN was diluted with F12 at a 4-fold dilution and inoculated into 96-well plates with 100 μL per well. The supernatant was incubated for 12 h and discarded, and the 96-well plates were inoculated with FAdV-4 at 100 TCID_50_. The cytopathic lesions were observed 24 h after infection, and the assay was calculated using the Reed–Muench method and expressed as UI/mg.

LMH cells were treated with 100–1000 ng of ChIFN-α or ChIFN-λ for 12 h and inoculated with 1 MOI of FAdV-4 to further compare the anti-FAdV-4 activity of ChIFN-α and ChIFN-λ. Cell samples were collected for Western blotting at 24 h and 48 h post-infection (hpi).

LMH cells were treated with 1000 UI of ChIFN-α or ChIFN-λ for 12 h and then inoculated with 1 MOI of FAdV-4 to elucidate the time frame of viral inhibition caused by ChIFN-α and ChIFN-λ. Cell and supernatant samples were collected at 12, 24, 36, 48, 60, and 72 hpi for further use.

### 4.5. Western Blot Analysis

Proteins (20 μg) were resolved using SDS-PAGE in Tris-Gly buffer at 150V and transferred to a polyvinylidene difluoride membrane (Sigma-Aldrich, St. Louis, Missouri, USA) at 250 mA. Specific mouse anti-hexon (1:8000 dilution with TBS) and rabbit anti-GAPDH monoclonal antibodies (1:10,000 dilution with TBS) were used for Western blotting. Protein expression was analyzed using a Tanon 5200 instrument (Tanon, Shanghai, China).

### 4.6. Virus Titration

LMH cells were inoculated into 96-well plates and allowed to attain 80% confluence to conduct the TCID_50_ assay. The samples underwent a 10-fold serial dilution at 100 µL/well, with three repetitions per sample. Following the manifestation of a noticeable cytopathic effect (CPE) in the cells, the number of wells exhibiting CPE at each dilution was documented, and the TCID_50_ value was determined using the Reed–Muench method.

### 4.7. Quantitative Real-Time PCR

mRNA was extracted from the cells using RNAfast2000 (Feijie, Shanghai, China) according to the manufacturer’s guidelines, and 100 ng of RNA was used for qRT-PCR experiments. Subsequently, HiScript II QRT SuperMix for quantitative PCR (Vazyme, Nanjing, China) was used for quantitative real-time PCR (qRT-PCR) analysis. The primer sequences used to quantify cDNA through qRT-PCR experiments were documented in Table 1. The relative quantities were determined using the 2^−ΔΔCT^ method and normalized to β-actin.

### 4.8. Statistical Analyses

Statistical analyses were conducted using GraphPad Prism 9.5.0 (GraphPad Software, San Diego, CA, USA), using paired *t*-tests and one-way ANOVA to assess group differences. *p*-values of <0.05 indicated statistical significance. The intensity of the protein bands was analyzed using Image J software (https://imagej.net/ij/; accessed on 11 December 2023; National Institutes of Health, Bethesda, MD, USA), and image layout and cropping were performed using Adobe Photoshop 2020 (Adobe, San Jose, CA, USA).

## 5. Conclusions

In conclusion, this study successfully generated ChIFN-α and ChIFN-λ through prokaryotic expression and demonstrated their strong anti-FAdV-4 activity in LMH cells. Furthermore, ChIFN-α and ChIFN-λ significantly enhanced *PKR*, *ZAP*, *IRF7*, *MX1*, *OASL*, *IFI6*, *IFIT5*, and *Viperin* expression levels. These findings suggest that ChIFN-α has stronger overall anti-FadV-4 activity than ChIFN-λ; however, ChIFN-λ exhibited greater efficacy in inhibiting FAdV-4 release during the early stages of infection. Our findings indicate that ChIFN-α and ChIFN-λ hold promise as potential therapeutic agents for treating HHS.

## Figures and Tables

**Figure 1 ijms-25-01681-f001:**
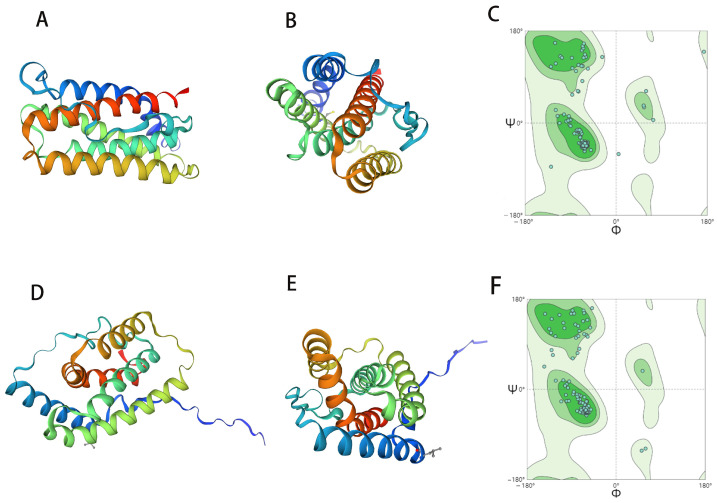
Structure prediction of chickem type I IFN-α (ChIFN-α) and type III IFN-λ (ChIFN-λ). The three-dimensional models of the ChIFN-α and ChIFN-λ proteins were reconstructed using the SWISS-MODEL online server (https://swissmodel.expasy.org/; accessed on 19 May 2023). Three-dimensional structure prediction of ChIFN-α (**A**,**B**). (**C**) shows the credibility of the prediction results, and the prediction results of one amino acid site are unstable. Three-dimensional structure prediction of ChIFN-λ (**D**,**E**). (**F**) shows the credibility of the prediction results.

**Figure 2 ijms-25-01681-f002:**
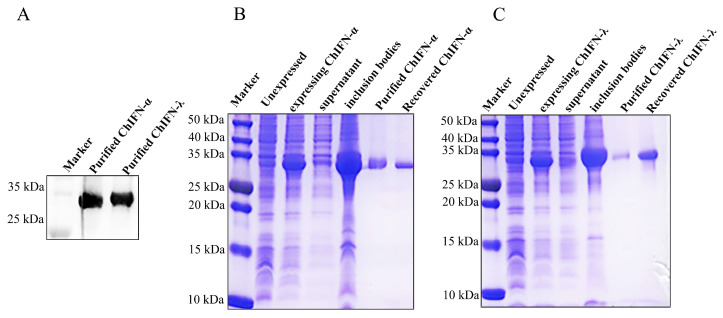
Purification and expression validation of ChIFN-α and ChIFN-λ. (**A**) Using Western blotting, 2 µg of the purified ChIFN-α and ChIFN-λ were validated. Validation of ChIFN-α (**B**) and ChIFN-λ (**C**) purification via sodium dodecyl-sulfate polyacrylamide gel electrophoresis (SDS-PAGE). ChIFN-α and ChIFN-λ were located in inclusion body expression and were purified using nickel columns with good purity.

**Figure 3 ijms-25-01681-f003:**
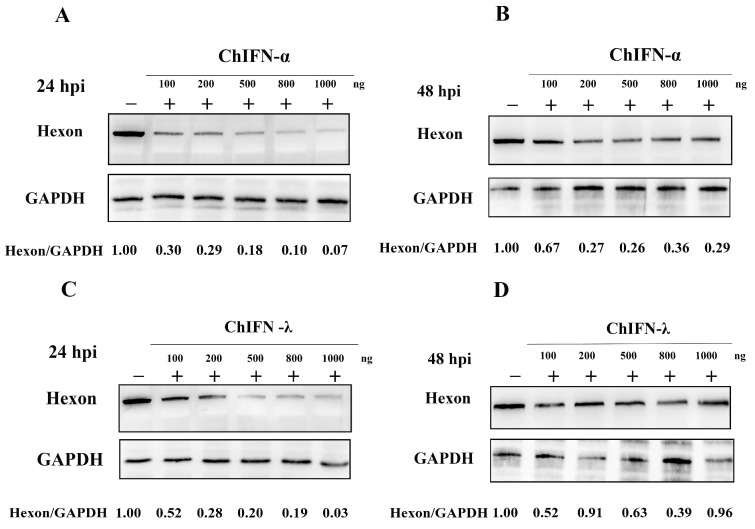
ChIFN-α and ChIFN-λ dose dependently inhibit fowl adenovirus serotype 4 (FAdV-4). (**A**) After pretreating Leghorn male hepatocellular (LMH) cells with different doses of ChIFN-α, the cells were inoculated with 1 MOI FAdV-4, and the inhibitory effect measured 24 h after inoculation increased with the ChIFN-α dose increase. (**B**) The ChIFN-α dose-dependent inhibition of FAdV-4 became insignificant after 48 h of inoculation. (**C**) ChIFN-λ pretreatment inhibited FAdV-4 dose dependently within 24 h after inoculation. (**D**) This dose-dependent inhibitory effect became non-significant 48 h after inoculation.

**Figure 4 ijms-25-01681-f004:**
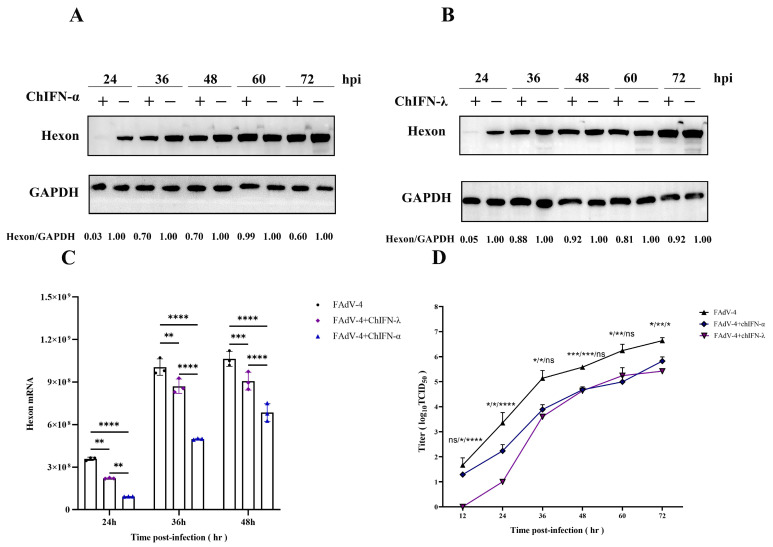
ChIFN-α (1000 UI) or ChIFN-λ (1000 UI) pretreatment inhibited hexon expression, transcription, and FAdV-4 viral release. After 12 h of ChIFN-α pretreatment, ChIFN-λ pretreatment, or no treatment, 1 MOI FAdV-4 was inoculated into the LMH cells. (**A**) and (**B**) Western blot experiments of protein samples collected at 12 h intervals after FAdV-4 inoculation. (**C**) Total mRNA was collected at 12 h intervals after FAdV-4 inoculation for RT-qPCR experiments. (**D**) One-step growth curves of interferon-treated and untreated groups. The first one shows the difference between FAdV-4 and ChIFN-α+FAdV-4, and the middle one shows the difference between FAdV-4 and ChIFN-λ+FAdV-4. The last one is the difference between ChIFN-α+FAdV-4 and ChIFN-λ+FAdV-4. The data above represent three independent experiments with three replicates in each experiment (the error bar represents the SEM) and were analyzed via one-way ANOVA using GraphPad Prism 9.5.0 software. (ns, not significant, * *p* < 0.5; ** *p* < 0.01; *** *p* < 0.001;**** *p* <0.0001 compared to the FAdV-4 group).

**Figure 5 ijms-25-01681-f005:**
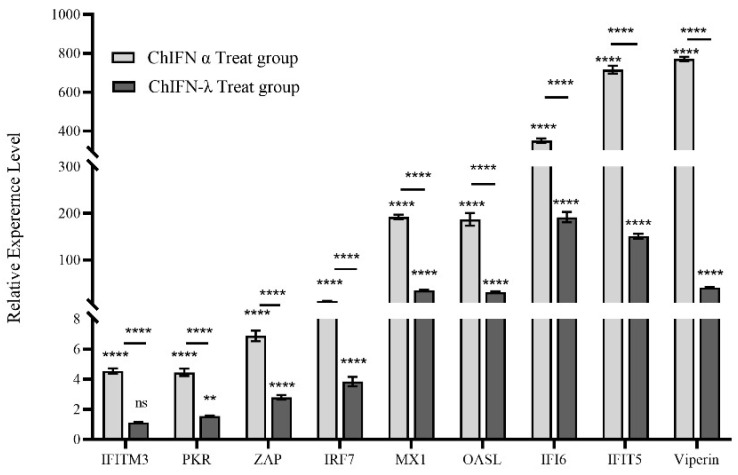
ChIFN-α and ChIFN-λ promoted mRNA expression of the ISGs *IFITM3*, *PKR*, *ZAP*, *IRF7*, *MX1*, *Viperin*, *IFIT5*, *OASL*, and *IFI6*. RT-qPCR experiments were performed on total mRNA collected 12 h after LMH cells were treated with or without ChIFN-α and ChIFN-λ. The relative quantities were determined using the 2^−ΔΔCT^ method and normalized to β-actin. The data above represent three independent experiments with three replicates in each experiment (the error bar represents the SEM) and were analyzed via one-way ANOVA using GraphPad Prism 9.5.0 software. (ns, *p* > 0.05; ** *p* < 0.01; **** *p* < 0.0001 compared to the negative control group).

**Table 1 ijms-25-01681-t001:** Primers for gene cloning and reverse transcription qPCR.

Name	Sequence(5′-3′)
Primers for gene cloning	
ChIFN-α-F	*CCGGAATT*CTGCAACCACCTTC
ChIFN-α-R	*CCCAAGCTT*CTAAGTGCGCGTGTTGCC
ChIFN-λ-F	*CCGGAATT*CCAGGTCACCCCGAAGAA
ChIFN-λ-R	*CCCAAGCTTC*TAAGTGCAATCCTCGCGCTGGGC
Primers for RT-qPCR	
Q-Hexon-F	CGAGGACTACGACGATTA
Q-Hexon-R	CGTGATACAGCAGGTTAATG
Q-MX1-F	AAGCCTGAGCATGAGCAGAA
Q-MX1-R	TCTCAGGCTGTCAACAAGATCAA
Q-OASL-F	ACATCCTCGCCATCATCGA
Q-OASL-R	GCGGACTGGTGATGCTGACT
Q-IFIT5-F	TGCTCTGAGGGAAGAACCCAACA
Q-IFIT5-R	AGGCTCCAGGGATGAGTCCACTT
Q-Viperin-F	AACGGTGGTTCAAGAAGTATGG
Q-Viperin-R	ACAGCATAATCTCGGCACCA
Q-IFITM3-F	TGGTGACGGTGGAGACG
Q-IFITM3-R	GGCAACCAGGGCGATGA
Q-ZAP-F	TTCCAAGTCAAGCCTGTCCC
Q-ZAP-R	CTCCGCTCTGCCTCTTCATC
Q-PKR-F	TGACTTCTGTGACATACAACCCTC
Q-PKR-R	TTTCAAACCAAATCAATCCC
Q-IRF7-F	AACGACGACCCGCACAAG
Q-IRF7-R	GCAGCAGGTCCAAATCCA
Q-IFI6-F	TCAACACACTCCTCAGGCTTTACC
Q-IFI6-R	GAACTCCGCCTCCGCAAGAG
Q-β-actin-F	CAACACAGTGCTGTCTGGTGGTA
Q-β-actin-R	ATCGTACTCCTGCTTGCTGATCC

## Data Availability

Data are contained within the article.

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
