# Peer review of "Chicken Interferon-Alpha and -Lambda Exhibit Antiviral Effects against Fowl Adenovirus Serotype 4 in Leghorn Male Hepatocellular Cells"

_ijms, 2024, doi:10.3390/ijms25031681_

Round 1

Reviewer 1 Report

Comments and Suggestions for Authors

Review of manuscript # IJMS-2794793 by Lai et al.

In this manuscript # IJMS2794793 submitted to IJMS by Lai et. al, the authors test and compare the anti-viral activity of ChIFN-alpha and gamma against Fowl adenovirus 4. The authors expressed ChIFN alpha and lambda in E. coli and treated LMH cells with purified IFN proteins for 12h followed by viral infection. From the data, the authors conclude that while IFN lambda has a significant impact on viral replication at early time point post infection, interferon alpha seems to have a more pronounced anti-viral effect at later time points. There was discordance between reduction in transcript and release of viral progeny. The manuscript is well written; however besides addressing the question asked leaves a lot of open questions unaddressed. Listed below are the main critiques of the manuscript.

1. Please show clearly in Figure 2D which data points in the growth curve are significantly different relative to the untreated wild type FAdv 4.0

2. Figure 5. Y- axis should be Relative Expression Level and not experence level. What was the rationale behind treating cells with 1000 UI of the IFNs? How was relative expression calculated ? How much RNA was used in the qRT-PCR analysis?

3. In Figure 4, please indicate the statistical significance in the graph plot area instead of below the X-axis where they are very hard to decipher. 

4. Experimental details for protein purification are poorly described and do not allow reproducibility. Please indicate O.D to which bacterial culture was grown, how much protein was loaded onto gels, running buffer and voltage, gel percentage etc.

5. In line 306 please indicate that supernatant was discarded not plates. Please state the catalog numbers of antibodies, the dilutions and the dilution buffer used for Western blotting.

6. Did the authors test the effect of ChIFN pretreatments with other MOIs of infection and other strains of FAdV ? Do they observe similar trends to indicate that these are not strain specific?

7. How do the authors speculate ChIFN treatment will be applied in a commercial setting? This is important since the authors state that likely degradation of IFNs during incubation affected their anti-viral activity. There is no data shown to this effect.

8. Please show that the LMH cells express ChIFN alpha and lambda receptors at the transcript and protein level.

9. Figure 2. C. It is not clear how the recovered ChIFN lambda band is more intense than the purified ChIFN lambda? how much protein was loaded on the gel. this is not a western blot but rather a SDS PAGE picture.

10. Figure 2C. Was the difference between the lambda and alpha significant at 36h time pi? Please run a 2- way ANOVA as well to determine this. State what p value is indicated by *****.

Comments on the Quality of English Language

English is sufficiently clear and easy to understand overall. 

Reviewer 2 Report

Comments and Suggestions for Authors

The manuscript describes effect of two types of chicken interferons (INF-alpha and IFN-lambda) on infection by fowl adenovirus FAdV-4. It expresses these interferons first in bacteria. Infections are performed on chicken hepatocellular LHM cells. Moderate antiviral effect is detected, and IFNs induce IFN-stimulated genes to various extent.

Comments:

1. There seem to be major flaws in experimental design. The IFNs are expressed in bacteria first, then IFN activity titered using FAdV-4 virus. At least that is shortly written in methods section. However, if studying effect of these IFNs on FAdV-4, titration should be done by some independent standard method, e.g. using VSV virus. The method section references (ref 52) and article that uses VSV, so it is very confusing. If adenovirus under study was used for titration, then the results cannot be interpreted.

2. Introduction is too general, it shloud not broadly cover the IFN system. It should focus more on known interaction of adenovirus with chicken IFN pathways (e.g. this reference missing PMID: 31874355)

3. Figure 1A - not necessary to predict chicken IFNA signal peptide. This protein has been produced in bacterial many times before, descriptions are available in the literature. Also, no clear description what was used for structure prediction (should be given in methods)

4. Figure 4 - legend should say what IFN dose was used

5. Figure 5 - not clear how significance was determined. E.g. if IRF7 is induced 12-fold, how comes that it is not significant?

Comments on the Quality of English Language

only minor edits necessary

Author Response

Dear Reviewer,

  Thank you very much for your valuable comments on our manuscript ijms-2794793. In accordance with your suggestions, we have implemented the following modifications to the manuscript. For these modifications we have adopted a yellow background for your review.

  1. There seem to be major flaws in experimental design. The IFNs are expressed in bacteria first, then IFN activity titered using FAdV-4 virus. At least that is shortly written in methods section. However, if studying effect of these IFNs on FAdV-4, titration should be done by some independent standard method, e.g. using VSV virus. The method section references (ref 52) and article that uses VSV, so it is very confusing. If adenovirus under study was used for titration, then the results cannot be interpreted.

Response:Thank you very much for your comments. I'm sorry we made the methods section too short. In fact, the biological activities of ChIFN-α and ChIFN-λ were assessed through the utilization of a standardized method(PMID: 22564876), specifically titration with vesicular stomatitis virus. This particular aspect of the experiment was conducted on DF-1. The determined activity of CHIFN-α against VSV was 2.58×105 IU/mg, while that of CHIFN-λ was found to be 5.3×104 IU/mg. We applied recombinant ChIFN-α and ChIFN-λ to anti-FADV-4 studies only after they were established to be biologically active.

Because it is more necessary to know the anti-FadV-4 activity of CHIFN-α and CHIFN-α, we did not present this part of the results in the manuscript. According to your suggestion, we have rewritten this part of the experiment in the methods(lines 317-318).

  1. Introduction is too general, it shloud not broadly cover the IFN system. It should focus more on known interaction of adenovirus with chicken IFN pathways (e.g. this reference missing PMID: 31874355)

Response: Thank you very much for your comments. We have removed the systematic introduction to interferon based on your suggestion(lines 50-54). However, we neglected the interaction of FAdV-4 with the interferon pathway. According to your suggestion, we have added the literature you suggested in lines 54-57.

  1. Figure 1A - not necessary to predict chicken IFNA signal peptide. This protein has been produced in bacterial many times before, descriptions are available in the literature. Also, no clear description what was used for structure prediction (should be given in methods)

Response: Thank you very much for your advice. We have removed the content of the signal peptide prediction according to your suggestion. The method of 3D structure prediction was added to lines 294-296.

  1. Figure 4 - legend should say what IFN dose was used

Response:It has been corrected in line 179 according to your comments.

  1. Figure 5 - not clear how significance was determined. E.g. if IRF7 is induced 12-fold, how comes that it is not significant?

Response:We apologize for the slight negligence in the processing of the data. We have re-analyzed the data and corrected the questions you raised (line 199).

Round 2

Reviewer 2 Report

Comments and Suggestions for Authors

The authors answered my review requests.

Comments on the Quality of English Language

acceptable